# Genome-wide CRISPR screening identifies Annexin A1 as a facilitator of porcine astrovirus entry

Yuhang Luo[1,2,3], Qingting Dong[4], Shiqin Yi[1,2,3], Wenting Zhang[1,2,3], Yiyang Du[1,2,3], Qingli Fang[5], Wenchao Zhang[6], Kang Ouyang[1,2,3], Ying Chen[1,2,3], Yeshi Yin[1,2,3], Zuzhang Wei[1,2,3], Yifeng Qin[1,2,3]*, Weijian Huang [1,2,3]*

1 Laboratory of Animal Infectious Diseases and Molecular Immunology, College of Animal Science and Technology, Guangxi University, Nanning, China, 2 Guangxi Key Laboratory of Animal Reproduction, Breeding and Disease Control, Nanning, China, 3 Guangxi Zhuang Autonomous Region Engineering Research Center of Veterinary Biologics, Nanning, China, 4 Guangxi Vocational University of Agriculture, Nanning, China, 5 Guangxi University Key Laboratory of Pathogenic Biology, Guilin Medical University, Guilin, China, 6 National Key Laboratory of Non-food Biomass Energy Technology, Guangxi Academy of Sciences, Nanning, China

* qinyf@gxu.edu.cn (YQ); huangweijian-1@163.com (WH)

## Abstract

Porcine astrovirus (PAstV) is an important and widespread pathogen in swine, linked to diarrheal outbreaks and extraintestinal disease. How PAstV enters host cells has remained unclear, and no cellular factor has been defined for PAstV entry. Here, a genome-wide CRISPR–Cas9 loss-of-function screen in porcine epithelial cells identifies Annexin A1 (ANXA1) as a host factor that facilitates PAstV entry. Genetic ablation or pharmacological/antibody blockade of ANXA1 reduces binding, lowers early viral RNA and capsid signals, and delays the rise of progeny, whereas re-expression restores susceptibility. Biochemical assays and surface plasmon resonance indicate a direct interaction between ANXA1 and the acidic C-terminal domain of the PAstV ORF2 capsid protein, and imaging shows ANXA1 co-localizes with incoming particles at the cell surface and supports attachment and uptake. Loss of ANXA1 does not alter infection by the non-astrovirus panel tested, indicating selectivity for PAstV under our conditions. Notably, infection is reduced but not abolished in ANXA1-deficient cells, consistent with additional entry factors acting alongside ANXA1. These findings position ANXA1 as an entry cofactor for PAstV and provide a mechanistic basis to refine models of astrovirus host-cell recognition.

## Author summary

Porcine astroviruses are common causes of diarrhea in piglets, yet how these viruses enter host cells is still unclear. Here, we used a genome-wide CRISPR–Cas9 knockout screen in porcine kidney cells to search for host genes

**Data availability statement:** All the data generated in this study are included in the S1 Data. The sequencing data generated in this study have been deposited in the NCBI Sequence Read Archive. The raw NGS data of CRISPR screening conducted in this study can be accessed through project PRJNA1293091; the raw RNA-seq data generated during this study are available under accession number PRJNA1295125.

**Funding:** This work was supported in part by National Natural Science Foundation of China (32260875 to W.J.H), Innovation Project of Guangxi Graduate Education (YCBZ2025005 to Y.H.L) and Guangxi Natural Science Foundation (2025GXNSFBA06955 to Q.T.D). The funders had no role in study design, data collection and analysis, decision to publish, or preparation of the manuscript.

**Competing interests:** The authors have declared that no competing interests exist.

that support porcine astrovirus type 1 (PAstV1) infection. We identified the membrane-associated protein Annexin A1 (ANXA1) as an entry cofactor. Cells lacking ANXA1 bound fewer viruses, showed reduced early viral RNA and capsid protein signals, and produced fewer progeny virions, whereas restoring or overexpressing ANXA1 increased susceptibility. Biochemical and imaging experiments indicate that ANXA1 directly binds the acidic C-terminal region of the viral capsid and co-localizes with incoming particles at the cell surface to support attachment and uptake. Loss of ANXA1 did not alter infection by several unrelated RNA viruses, suggesting selectivity for porcine astroviruses under our conditions. We further show that ANXA1 influences host responses, contributing to RIG-I–IRF3 signalling and virus-induced apoptosis. Pharmacological or antibody-mediated targeting of ANXA1 lowered PAstV replication in cell culture, highlighting the virus–ANXA1 interface as a potential, though context-dependent, target for antiviral intervention.

## Introduction

Astroviruses (AstVs), belonging to the family *Astroviridae*, are non-enveloped, positive-sense, single-stranded RNA viruses causing gastrointestinal and extraintestinal diseases in a wide range of mammalian and avian hosts [1,2]. Porcine astroviruses (PAstVs), classified into five genotypes (PAstV1–PAstV5) based on capsid protein sequence variability, are globally prevalent in swine populations, frequently detected in diarrheal piglets, and commonly associated with mild-to-moderate enteric diseases, villous atrophy, intestinal barrier disruption, and transient growth retardation [3,4]. Recent epidemiological studies have indicated high infection rates of PAstVs in pig herds worldwide, underscoring their significant economic impact on swine production [5–7]. Despite this widespread distribution and clear pathogenicity, the molecular mechanisms underpinning PAstV infection remain largely unknown, and a definitive cellular receptor mediating PAstV entry has not yet been identified. Notably, neonatal Fc receptor (FcRn) and dipeptidyl peptidase 4 (DPP4) were recently reported as cellular entry factors for human astrovirus (HAstV) [8–11]; however, whether analogous receptors or similar molecular mechanisms are exploited by PAstV remains to be elucidated.

Annexin A1 (ANXA1) is a multifunctional calcium-dependent phospholipid-binding protein widely recognized for its roles in inflammation, immune modulation, and cellular signaling pathways [12,13]. Structurally, ANXA1 possesses a conserved C-terminal core domain and a distinct N-terminal region responsible for mediating interactions with cellular membranes and regulatory proteins [14]. Recent studies have implicated ANXA1 in various viral infections [15], including herpes simplex virus 1 (HSV-1), influenza A virus (IAV), and foot-and-mouth disease virus (FMDV). For instance, suppression of ANXA1 significantly reduces HSV-1 lethality in infected mice, highlighting its role in promoting viral pathogenicity. During FMDV infection, ANXA1 expression is markedly increased, enhancing type I interferon (IFN-I)

responses via the TBK1–IRF3 signaling pathway; however, the FMDV 3A protein directly interacts with ANXA1, antagonizing its antiviral function and thus facilitating viral replication [16–18]. Similarly, in IAV infection, ANXA1 promotes virus entry and nuclear trafficking, thereby increasing viral load despite concurrently activating the RIG-I-mediated antiviral signaling pathway [19]. Nevertheless, the precise role and molecular mechanisms underlying ANXA1 as a host factor facilitating viral replication and entry, remain poorly defined and require further investigation.

In this study, we used a genome-wide CRISPR screen to identify ANXA1 as a host factor that facilitates PAstV entry [20,21]. ANXA1 supports attachment through direct interaction with the capsid protein open reading frames2 (ORF2), involving the acidic C-terminal domain of the capsid and the Repeat III region of ANXA1. Deletion of ANXA1 lowered binding, reduced early viral RNA and capsid signals, and delayed replication, yet infection was not abolished, indicating that additional entry factors likely contribute. ANXA1 also modulates host responses: loss of ANXA1 was associated with reduced apoptosis and a dampened RIG-I–IRF3 pathway, whereas re-expression restored these readouts. Pharmacological or antibody blockade of ANXA1 decreased PAstV readouts in susceptible cells. Together, these findings indicate that ANXA1 acts as an entry cofactor and immunomodulator in PAstV infection, refining current models of astrovirus–host interactions and pointing to the virus–ANXA1 interface as a potential point for intervention.

## Results

### CRISPR screening identifies ANXA1 as a entry factor for PAstV infection

To identify host factors essential for cell survival upon PAstV infection, we initially evaluated the replication efficiency and cytopathic effects of PAstV (Mamastrovirus 3 isolate PAstV-GX1) in different cell lines. Compared to IPEC-J2 cells, PK15 cells exhibited significantly higher viral titers and more pronounced cytopathic effects (S1A Fig). In PK-15 cells, PAstV at 0.01 MOI caused gradual cell death by 72 h post-infection. MOIs > 0.01 produced complete cytopathic effect by 24–48 h, whereas 0.001 MOI was too slow; thus 0.01 MOI was used for screening (S1B Fig). Subsequently, PK15 cells stably expressing Cas9 were transduced with a lentiviral library containing 125,951 single-guide RNAs (sgRNAs) targeting 41,983 genes, followed by puromycin selection, resulting in a highly uniform knockout (KO) library with comprehensive coverage. A positive-selection screening approach involving four consecutive rounds of PAstV infection was then performed to identify host factors critical for PAstV infection (Fig 1A and 1B) shows the first-round selection. PK15 and PK15Cas9 were infected with PAstV at MOI 0.01 and imaged at 96 h. Wild-type (WT) PK15 showed complete CPE with no surviving cells. PK15-Cas9 retained small patches of attached cells. Surviving PK15Cas9 cells were carried forward for the next infection.

A total of 123 candidate host factors were identified (absolute log2 fold change ≥1.5, MAGeCK RRA P < 0.05, at least one active sgRNA). For prioritization, hits were ranked by RRA score and filtered for support from multiple sgRNAs or high sgRNA-consistency scores, stable positive effects across the two selection cycles sampled from the same screen (post-round 2 and post-round 4 collections), no growth effect in mock-selected controls, and not annotated as core-essential. We then applied biological plausibility filters, favouring genes with membrane or endocytic annotations and with detectable expression in PK15. ANXA1 met these criteria and ranked among the top hits (Fig 1c–1d). We therefore advanced ANXA1 and other leading candidates to siRNA validation, including AKT serine/threonine kinase 2 (AKT2), calmegin (CLGN), iron responsive element binding protein 2 (IREB2), LIM homeobox transcription factor 1 beta (LMX1B), phosphatase and tensin homolog (PTEN), BRICK1 subunit of SCAR/WAVE actin nucleating complex (BRK1), Janus kinase 2 (JAK2), symplekin scaffold protein (SYMPK), apoptotic peptidase activating factor 1 (APAF1), stress associated endoplasmic reticulum protein 1 (SERP1), general transcription factor IIE subunit 1 (GTF2E1), and ANXA1, with knockdown efficiencies validated via qPCR (S1C Fig). Subsequently, siRNA-treated cells were infected with PAstV at an MOI of 1, and infection levels were assessed through immunostaining. Knockdown of CLGN, JAK2, APAF1, and notably ANXA1 markedly decreased viral infection (Figs 1E and S1D). As ANXA1 siRNA had the most pronounced impact in the screen, we chose ANXA1 for deeper investigatio.

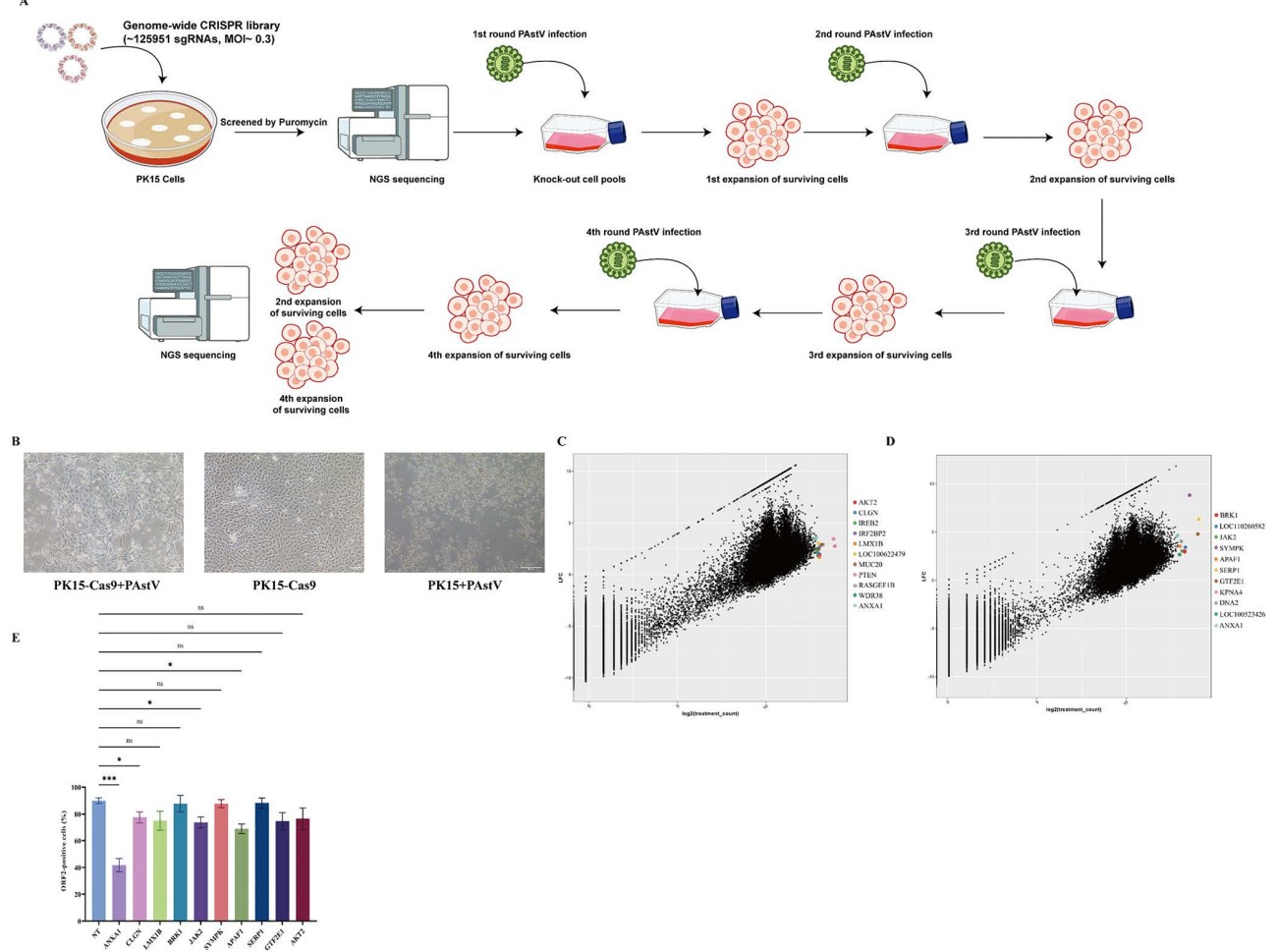

**Fig 1. CRISPR screening identifies ANXA1 as a entry factor for PAstV infection. (A)** Schematic of genome-wide CRISPR screen for host factors required for PAstV infection. Illustration from NIAID NIH BioArt Source (//bioart.niaid.nih.gov). PK15 cells were transduced with a CRISPR knockout library and subjected to four rounds of PAstV infection. Surviving cells from rounds 2 and 4 were harvested and analyzed by sequencing (n = 2 biological replicates). **(B)** PK15 and PK15-Cas9 were infected at MOI 0.01 and imaged at 96 h. **(C, D)**, Scatterplots depicting sgRNA enrichment in rounds 2 **(C)** and 4 **(D)** of PAstV screening. **(E)** Percentage of PAstV-infected cells (MOI = 1, 24 hpi) in indicated knockdown lines, assessed by immunostaining with mouse anti-Capsid antibody. Data are normalized to non-targeting (NT) controls and presented as mean ± SD (n = 4 independent experiments). Data represent mean ± SD (n = 3). Statistical significance by unpaired two-tailed Student's t-test. (ns, P > 0.05; *P < 0.05; **P < 0.01; ***P < 0.001).

## ANXA1 supports PAstV infection

ANXA1 is a cytoplasmic protein known to externalize and localize at the cell surface [16,22,23]. To initially characterize ANXA1 expression, we performed flow cytometry analysis on PK15 and IPEC-J2 cells, confirming robust expression in both cell lines (Fig 2A). We then utilized the CRISPR/Cas9 system to successfully generate ANXA1 knockout (KO) PK15 cell lines, employing specific sgRNAs targeting exon 4 of the ANXA1 gene. Several single-cell-derived KO clones were confirmed by DNA sequencing to harbor precise genetic edits without off-target mutations. Importantly, these KO cell lines displayed similar growth kinetics compared to WT cells, excluding potential interference from altered cell viability upon subsequent viral challenge (Figs 2B-2C and S2A-S2B). As expected, ANXA1-KO clones demonstrated significant resistance to PAstV infection compared to WT cells (S2C Fig). Consistently, viral RNA and protein levels were substantially

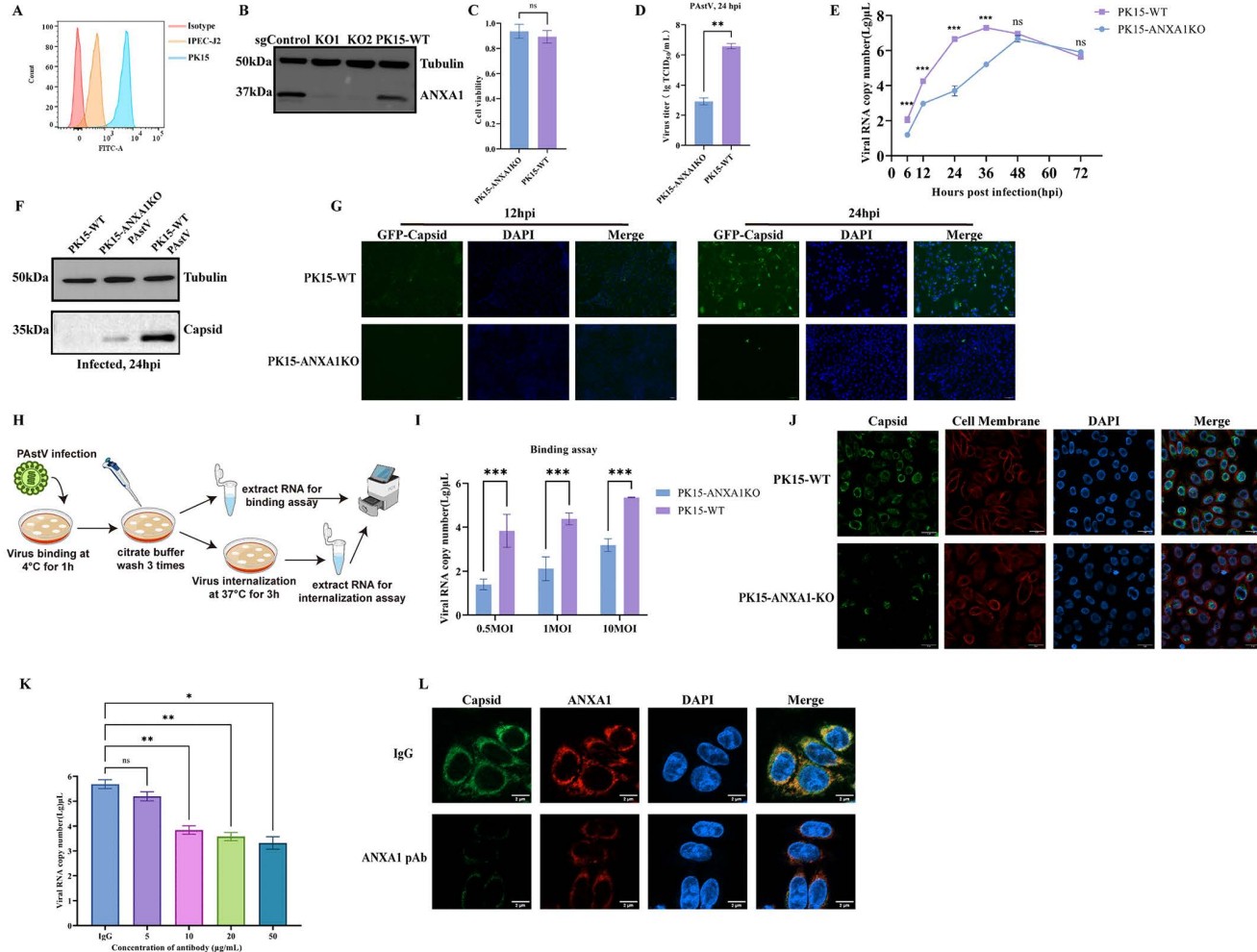

**Fig 2. ANXA1 supports PAstV infection. (A)** Representative histograms of ANXA1 surface expression measured by flow cytometry. **(B)** Western blot confirming ANXA1 knockout (KO) in PK15 cells transduced with sgRNAs (n = 2). **(C)** Viability of ANXA1 KO PK15 cells assessed by CCK8 assay. Data represent mean ± SD (n = 3). **(D)** Viral titers in WT and ANXA1 KO PK15 at 24 hpi after infection at MOI 0.01. **(E)** Viral RNA in WT and ANXA1 KO PK15 at the indicated times after infection at MOI 0.01. **(F)** Viral proteins in WT and ANXA1 KO PK15 at 24 hpi after infection at MOI 0.01, by Western blot. **(G)** Immunofluorescence detection of PAstV infection (MOI = 0.01) in WT and ANXA1 KO PK15 cells. Nuclei stained with DAPI; virus stained with mouse anti-PAstV Capsid antibody. Scale bar, 50 μm. **(H)** Overview of virus binding and internalization assay. Illustration from NIAID NIH BioArt Source (//bioart. niaid.nih.gov). **(I)** Impact of ANXA1 deletion on PAstV binding to PK15 cells. **(J)** Confocal imaging of virus particles (MOI = 10) binding to WT and ANXA1 KO PK15 cells. Scale bar, 5 μm. **(K)** Competitive blocking of PAstV binding with ANXA1 polyclonal antibody (pAb). PK15 cells were blocked with ANXA1 pAb (5–50 μg/ml) for 1 h and incubated with PAstV (MOI = 10) at 4°C for 1 h, binding quantified by RT-qPCR. **(L)** Confocal images showing PAstV binding (MOI = 10) after blocking with ANXA1 pAb (50 μg/ml) or mouse IgG (50 μg/ml). Scale bar, 5 μm. Data represent mean ± SD (n = 3). Statistical significance by unpaired two-tailed Student's t-test and Two-way ANOVA. (ns, $P > 0.05$; $*P < 0.05$; $**P < 0.01$; $***P < 0.001$).

reduced in ANXA1-KO cells relative to WT controls (Fig 2D-2F). The growth curve showed significant reductions in KO from 6 to 36 hpi, with curves approaching each other after 48 hpi (Fig 2E). Immunofluorescence at 12 and 24 hpi showed fewer Capsid-positive cells and weaker signal in KO, while WT showed widespread infection (Fig 2G). These early differences indicate that ANXA1 acts at the start of infection. Given that ANXA1 serves as a membrane-associated receptor mediating attachment and internalization for multiple viruses, we hypothesized its involvement at the viral entry stage [16,17,24]. To test this hypothesis, we assessed virus binding in WT and ANXA1-KO cells exposed to varying MOI

(Fig 2H). Virus binding was markedly diminished in ANXA1-KO cells compared to WT counterparts; after the 37°C shift, the total internalized genomes per well were lower in ANXA1-KO, while internalization efficiency, calculated as internalized divided by bound, was similar (Figs 2I and S2D). Confocal microscopy visualization provided additional direct evidence supporting ANXA1-mediated enhancement of viral attachment at the cell surface (Fig 3j). To further confirm the critical role of ANXA1 during early PAstV infection, we performed a series of competitive binding assays at both cellular and viral levels [25]. PK15 cells were pretreated for 1 h at 37°C with ANXA1 polyclonal antibody at 5–50 μg/mL or mouse IgG, then exposed to PAstV for 1 h at 4°C to measure binding. qPCR showed a dose-dependent drop in binding with ANXA1 antibody (Fig 2K). Confocal imaging confirmed fewer surface virions after blocking with 50 μg/mL ANXA1 antibody, while mouse IgG had little effect (Fig 2I). Purified recombinant ANXA1 is shown in S2F Fig. Collectively, these findings provide robust preliminary evidence identifying ANXA1 as a host factor facilitating PAstV entry into host cells.

## ANXA1 facilitates early PAstV entry and rescue restores susceptibility

Our preliminary results indicated that ANXA1 functions as a host factor mediating PAstV entry. To further explore whether increased ANXA1 expression could enhance PAstV entry, we constructed ANXA1-overexpressing plasmids and transiently transfected these into PK15 and IPEC-J2 cells. Western blot analysis confirmed successful ANXA1 overexpression in both cell lines (Fig 3A). Subsequent PAstV infection demonstrated that ANXA1 overexpression significantly enhanced viral entry in both PK15 and IPEC-J2 cells (Fig 3B).

To further validate the requirement of ANXA1 in PAstV replication, we performed rescue experiments. We amplified full-length ANXA1 from PK15 genomic DNA and introduced synonymous mutations within the PAM sequence to render it resistant to CRISPR/Cas9 cleavage [26], verified by DNA sequencing (Figs 3C and S2E). This modified ANXA1 sequence was then cloned into a lentiviral vector and introduced into the ANXA1-KO cells, generating stable cell lines designated A1KO-rescue. Western blot confirmed complete restoration of ANXA1 expression in these cells (Fig 3D). qRT-PCR analysis demonstrated that reintroduction of ANXA1 restored susceptibility to PAstV infection to levels comparable to WT cells, confirming ANXA1 as an essential host factor (Fig 3E).

Given that IPEC-J2 cells are also susceptible to PAstV infection, we extended our investigation by establishing ANXA1-knockout polyclonal IPEC-J2 cells [27]. Immunofluorescence staining (S2F Fig) and qRT-PCR assays (Fig 3F) confirmed that PAstV replication was significantly reduced in ANXA1-KO IPEC-J2 cells compared to WT counterparts. To evaluate specificity, we infected PK15-WT and PK15-ANXA1KO with Getah virus (GETV), Enterovirus G (EVG), Seneca Valley virus (SVV), Mammalian orthoreovirus (MRV), Bovine Enterovirus (BEV), and PAstV genotype 5. Immunofluorescence at 24 h showed similar replication for GETV, EVG, SVV, MRV, and BEV in WT and KO, whereas PAstV genotype 5 was reduced in KO (Fig 3G). RT-qPCR at 24 h after infection at MOI 0.1 confirmed no significant differences for the five non-PAstV viruses and lower PAstV genotype 5 RNA in KO (Fig 3H). These data indicate that ANXA1 specifically supports PAstV replication. Additionally, to assess whether PAstV infection impacts ANXA1 expression, PK15 cells infected with PAstV at an MOI of 0.1 were analyzed. The results showed a gradual increase in ANXA1 expression following PAstV infection (Fig 3I-3J).

## ANXA1 binds to the PAstV Capsid acidic domain directly via its repat III domain

To investigate the physical interaction between ANXA1 and PAstV proteins, we first confirmed expression of PAstV proteins Capsid, ns1a1, ns1a3, and ns1a4 in HEK293T cells (S3A Fig). Co-immunoprecipitation (co-IP) assays revealed that only Capsid, but none of the other tested proteins, interacted with ANXA1 (Fig 4A). Confocal imaging of HEK293T cells co-expressing ANXA1 and PAstV Capsid showed colocalization. ImageJ analysis gave a Pearson's correlation coefficient of 0.68, consistent with overlapping red and green line profiles (Fig 4B), which was further validated by bimolecular fluorescence complementation (BiFC) assay, confirming a direct interaction between the two proteins (S3B Fig). Next, we employed AlphaFold3 to predict the 3D structures of ANXA1 and Capsid (S3C Fig). Protein-protein docking using the local

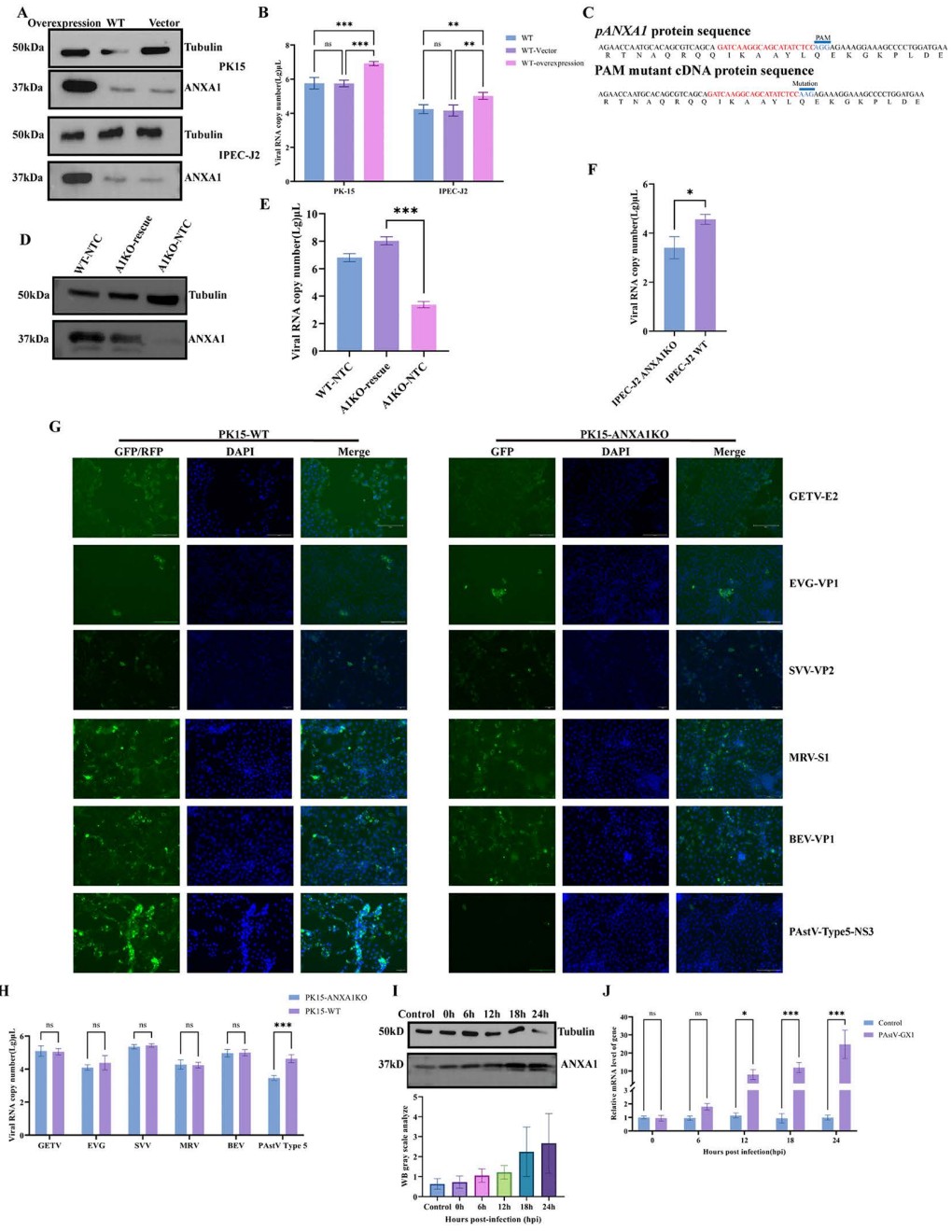

**Fig 3. ANXA1 facilitates early PAstV entry and rescue restores susceptibility. (A)** Western blot validating ANXA1 overexpression in PK15 and IPEC-J2 cells using mouse anti-ANXA1 antibody. **(B)** RT-qPCR analysis of PAstV infection (MOI = 0.01) at 24 h in ANXA1-overexpressing PK15 and IPEC-J2 cells. **(C)** Schematic design of CRISPR-resistant ANXA1 (pANXA1). **(D)** Western blot confirming ANXA1 restoration in PK15-ANXA1KO after PAstV infection. **(E)** RT-qPCR analysis of PAstV infection (MOI = 0.01) at 24 h in ANXA1-rescued PK15 cells. **(F)** RT-qPCR analysis of PAstV infection (MOI = 0.01) at 24 h in ANXA1 polyclonal KO IPEC-J2 cells. **(G)** Immunofluorescence detection of virus replication in PK15-WT and PK15-ANXA1 KO cells. **(H)** RT-qPCR analysis of virus replication (MOI = 0.1) in PK15-ANXA1 KO cells at 24 h post-infection. **(I, J)** Western blot **(I)** and RT-qPCR **(J)** analysis of ANXA1 expression in PK15 cells infected with PAstV (MOI = 1). Data represent mean ± SD (n = 3). Statistical significance by unpaired two-tailed Student's t-test and Two-way ANOVA. (ns, P > 0.05; *P < 0.05; **P < 0.01; ***P < 0.001).

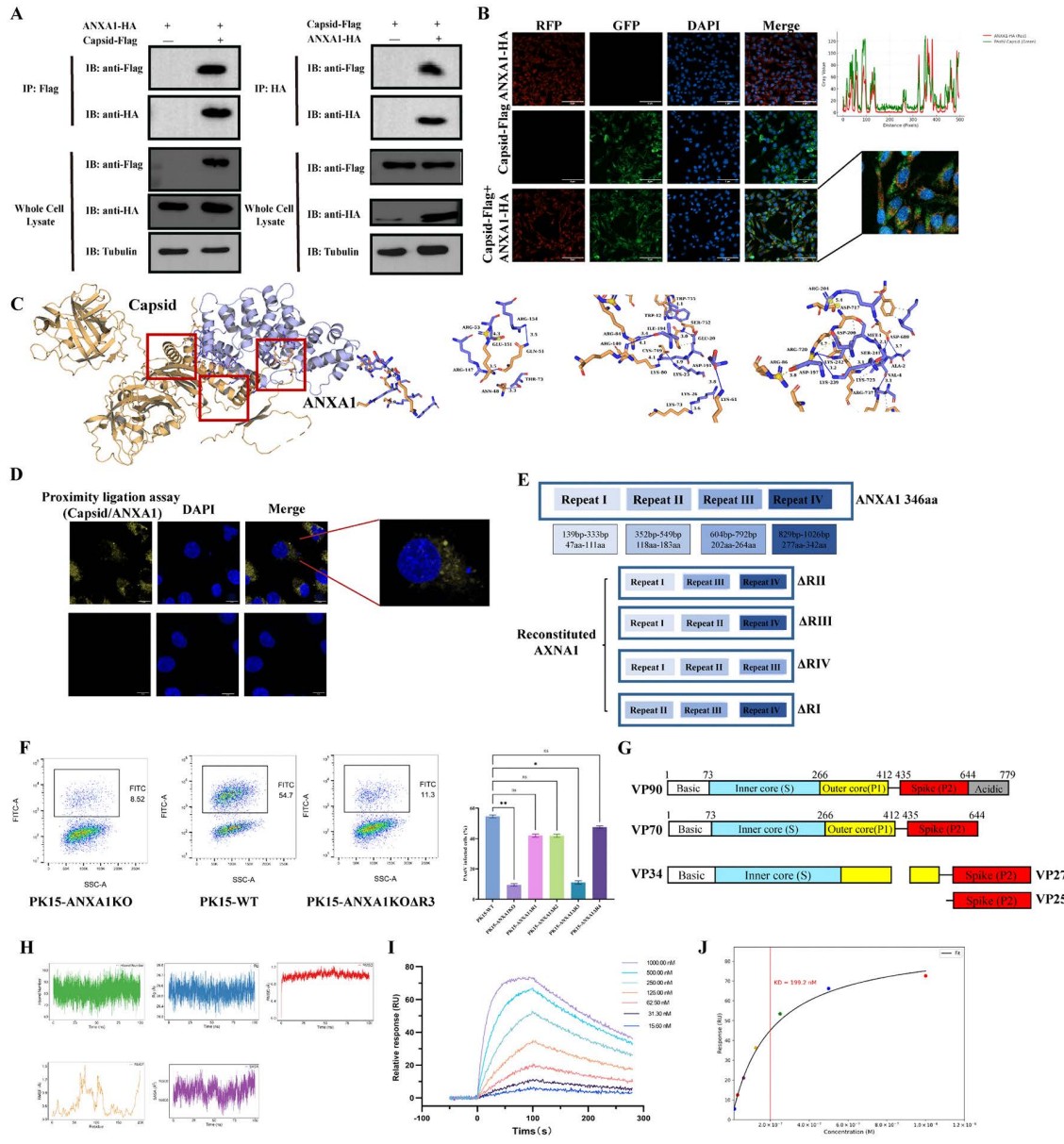

**Fig 4. ANXA1 binds directly to the PAstV Capsid acidic domain via its repeat III domain. (A)** Co-immunoprecipitation assay of ANXA1-HA and Capsid -Flag co-transfected in 293T cells, followed by Western blot detection (n = 2). **(B)** Confocal microscopy showing colocalization of ANXA1-HA and Capsid-Flag. Scale bar, 5 μm. **(C)** Predicted interaction interface between ANXA1 and Capsid by protein-protein docking using HDOCKlite v1.1. **(D)** PLA signal (orange) showing interaction between ANXA1 and Capsid in PK15 cells infected with PAstV (MOI = 1) for 24 **h.** Scale bar, 5 μm. **(E)** Schematic of ANXA1 conserved C-terminal domain composed of four annexin repeats, recombinant ANXA1 plasmids with HA-tag. **(F)** Flow cytometry analysis of PAstV-ILOV infection (MOI = 1) in PK15-ANXA1 KO cells transfected with different ANXA1 recombinant plasmids. Data represent mean ± SD (n = 3). Statistical analysis by unpaired two-tailed Student's t-test (ns, P > 0.05; *P < 0.05; **P < 0.01; ***P < 0.001). **(G)** Schematic depicting PAstV VP90 maturation process. **(H)** Molecular dynamics simulation of ANXA1-ΔR3 and Capsid acidic domain. **(I, J)** SPR analysis of ANXA1-ΔR3 binding to Capsid-acidic domain.

version of HDOCKlite v1.1 indicated a putative interaction interface between ANXA1 and Capsid, and subsequent MM/GBSA calculations performed by the HawkDock server predicted a binding free energy of -8.19 kcal/mol, further supporting their direct interaction [28–30] (Fig 4C). Having confirmed an exogenous interaction, we next asked whether ANXA1 and Capsid also interact endogenously in PK15 cells during viral infection. Employing a proximity ligation assay (PLA) [31], we observed endogenous interaction between ANXA1 and PAstV Capsid upon PAstV infection (Fig 4D).

Sequence homology analysis revealed that porcine ANXA1 shares 93.64% and 90.78% amino acid sequence identity with human and mouse orthologs, respectively (S3D Fig). All annexins possess a conserved C-terminal core composed of four annexin repeats (Repeat I to IV, except eight in ANXA6), each harboring characteristic type II calcium-binding sites formed by hydrophobic interactions that maintain structural stability [32,33]. In contrast, the unique N-terminal domain typically mediates interactions between annexins and specific intracellular protein partners, such as members of the S100 protein family [34]. To identify the ANXA1 domain responsible for Capsid interaction, we generated several deletion mutants fused to transmembrane and cytoplasmic domains (Figs 4E and S3E), transiently transfected these constructs into ANXA1-KO PK-15 cells, and then infected them with recombinant PAstV-ILOV virus previously constructed in our lab [35]. Flow cytometry analysis indicated that mutants lacking the Repeat III (R3) domain failed to fully restore viral infection, whereas mutants retaining this domain recovered viral infectivity (Figs 4F and S3F).

The PAstV genome (~7 kb) contains three ORFs, with ORF2 encoding the structural precursor capsid protein VP90 [1]. VP90 can be divided into three linear domains: a conserved N-terminal domain, a highly variable region, and an acidic C-terminal domain [9] (Fig 4G). Maturation of VP90 produces mature viral structural components: capsid protein VP34 and spike protein VP25/27. Based on previous docking predictions, we hypothesized that the acidic C-terminal domain of VP90 specifically interacts with the Repeat III domain of ANXA1. To further explore this interaction, we conducted molecular dynamics simulations, evaluating system stability via root-mean-square deviation (RMSD) and radius of gyration (RG), fluctuations of key residues through root-mean-square fluctuations (RMSF), solvent-accessible surface area (SASA), and calculated binding free energies using the MM/PBSA method. All these metrics consistently indicated a highly stable interaction characterized by constrained local dynamics, sustained hydrophobic interactions, stable hydrogen bond networks, and thermodynamically spontaneous binding (Fig 4H). Finally, to conclusively demonstrate a direct interaction, we expressed and purified the relevant protein domains and performed surface plasmon resonance (SPR) experiments. SPR analyses revealed that the acidic C-terminal domain of PAstV Capsid bound directly and with high affinity (KD = 199.2 nM) to ANXA1 Repeat III domain (Fig 4I-4J and S1 Table). Collectively, these data firmly establish that ANXA1 directly mediates PAstV entry through its Repeat III domain by binding specifically to the acidic C-terminal domain of the viral Capsid protein.

## ANXA1 modulates apoptosis and orchestrates RIG-I–mediated antiviral signaling during PAstV infection

Previous studies from our group demonstrated that PAstV infection induces caspase-3-dependent apoptosis, which is beneficial for viral replication. Given that ANXA1 is also implicated in promoting apoptosis, we assessed its role in PAstV-induced cell death [17,36]. After infecting PK15 cells with PAstV at an MOI of 0.1, ANXA1-KO cells exhibited substantially lower levels of apoptosis compared to WT cells (Fig 5A). Annexin V/PI staining further confirmed this observation, showing significantly higher early apoptotic cells in WT (29.7% ANXV+PI-) than in ANXA1-KO cells (13.5% ANXV+PI-) (Fig 5B). To minimise the confounding effect of reduced PAstV infection in ANXA1-KO cells, we used BEV, for which ANXA1 does not alter replication, and confirmed that BEV loads were comparable between WT and KO, whereas ANXV+PI- cells were still reduced in the KO (S4A Fig.) These results suggest that ANXA1 deficiency suppresses PAstV-induced apoptosis, potentially contributing to reduced viral replication.

Annexins play diverse roles, including extracellular activities associated with inflammation and fibrosis, as well as intracellular functions such as membrane repair, cytoskeletal remodeling, organelle trafficking, and signal transduction [37]. To elucidate the relationship between ANXA1 and PAstV-induced immune and inflammatory responses, we performed RNA

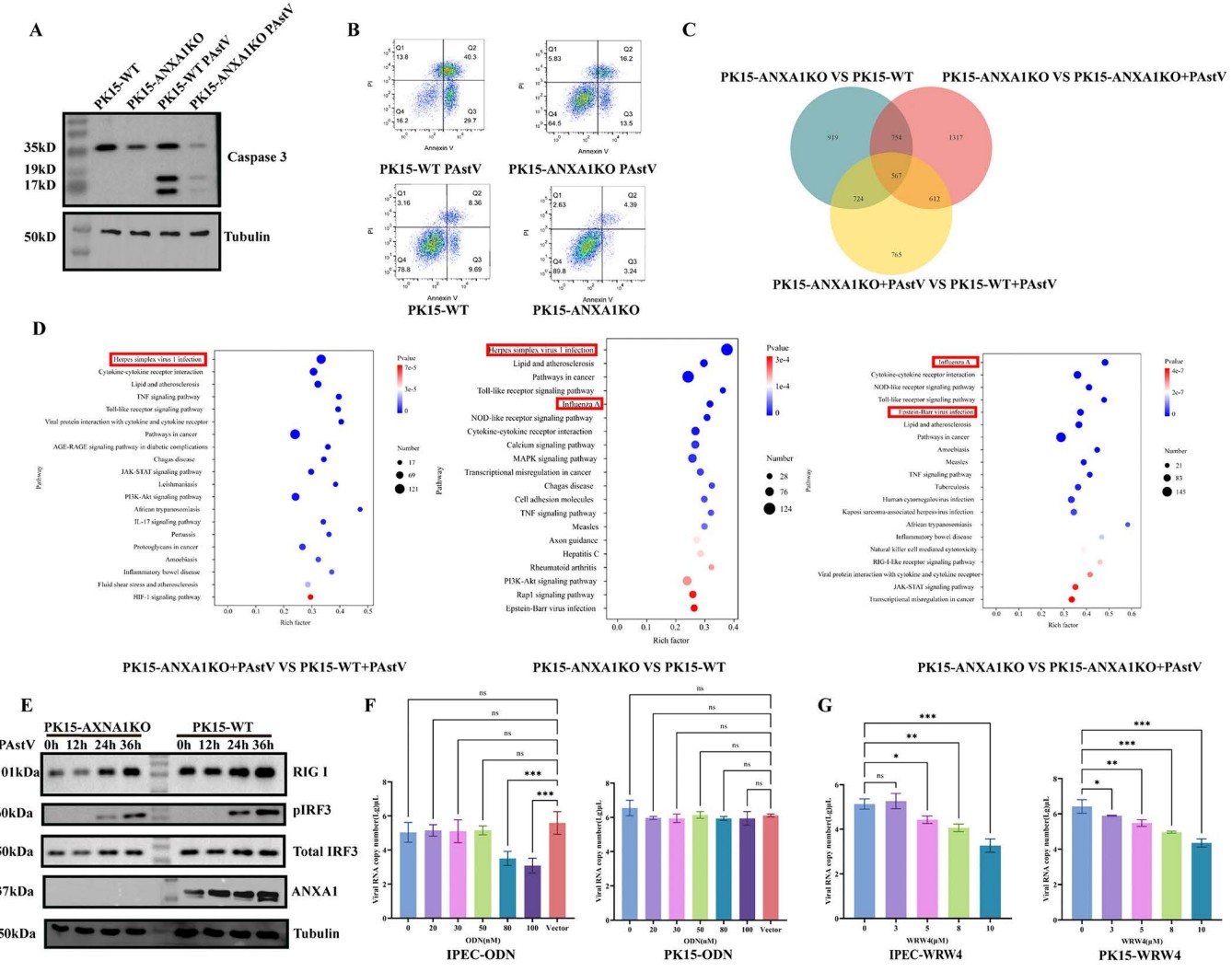

**Fig 5. ANXA1 modulates apoptosis, orchestrates RIG-I-mediated antiviral signaling, and pharmacological targeting of ANXA1 impairs PAstV infection ex vivo.** WT and ANXA1-KO PK15 cells infected with PAstV (MOI = 0.01). At 12 hpi, cell lysates were analyzed by Western blot for cleaved caspase-3 (**A**) or by Annexin V/PI staining (**B**) for apoptosis. (**C**) Venn diagram showing differentially expressed genes between PK15-WT and PK15-ANXA1 KO cells. (**D**) KEGG pathway analysis of differentially expressed genes. (**E**) Western blot of indicated proteins at specified time points (n = 2). (**F, G**) RT-qPCR analysis of PAstV infection (MOI = 0.01, 24 h) in PK15 and IPEC-J2 cells after ODN transfection (**F**) or WRW4 treatment (**G**). Data represent mean ± SD (n = 3). Statistical analysis by unpaired two-tailed Student's t-test (ns, P > 0.05; *P < 0.05; **P < 0.01; ***P < 0.001).

sequencing (RNA-seq) to identify global changes in cytokine signaling pathways and antiviral immune responses following viral infection. Approximately 510.6 million high-quality reads were generated, with sequence alignment and gene annotation achieving robust quality metrics. Gene expression distribution was highly consistent among replicates within each experimental group, as confirmed by principal component analysis (PCA) and Pearson correlation coefficients (S2-S3 Tables and S4B-S3D Figs). Differential expression analysis using DESeq2 software identified 567 genes commonly dysregulated across three comparisons. Specifically, we found 2,668 differentially expressed genes (DEGs) between PAstV-infected WT and ANXA1-KO PK15 cells, of which 866 were upregulated and 1,802 were downregulated in WT cells. Conversely, PAstV-infected ANXA1-KO cells exhibited a distinct expression profile, with 3,250 DEGs, including 2,113 upregulated and 1,137 downregulated genes (Figs 5C and S4E-S4F). Gene Ontology (GO) enrichment analysis revealed

that these DEGs predominantly belonged to biological processes associated with stimulus-response regulation, stress responses, and immune system functions (S4G Fig). Kyoto Encyclopedia of Genes and Genomes (KEGG) pathway analysis identified significant enrichment in the RIG-I-like receptor signaling pathway, consistent with previous findings from our laboratory (Fig 5D).

Protein-protein interaction (PPI) network analysis highlighted key regulatory nodes such as IL1A, TNF, and NFκBIA, closely associated with RIG-I signaling, indicating their potential roles in antiviral defense and possible targets for viral immune evasion strategies (S4H Fig). Notably, RNA-seq analysis revealed reduced RIG-I expression in ANXA1-KO cells compared to WT cells, a finding consistent with previous reports demonstrating ANXA1-mediated enhancement of RIG-I-dependent signaling via the IRF3-IFNAR-STAT1-IFIT1 pathway in A549 cells [19,38]. To further assess the involvement of ANXA1 in PAstV-triggered RIG-I signalling, we first bypassed viral entry and directly activated the pathway with 5′ppp-dsRNA. In the absence of infection, 5′ppp-dsRNA induced robust upregulation of phosphorylation of IRF3 in WT PK15 cells, whereas both p-IRF3 protein levels were clearly blunted in ANXA1-KO cells, indicating that ANXA1 contributes to RIG-I–IRF3 activation independently of PAstV entry(S4G Fig). Consistent with this, during PAstV infection the induction of RIG-I, p-IRF3 and total IRF3 over time was attenuated in ANXA1-KO cells compared with WT cells (Fig 5E), supporting a role for ANXA1 in sustaining RIG-I–IRF3 signalling during PAstV infection.

### Pharmacological targeting of ANXA1 impairs PAstV infection in cells ex vivo

Given reports that antisense oligodeoxynucleotides (ODN) targeting ANXA1 effectively suppress its expression in vitro [39,40], and that WRW4 (Trp-Arg-Trp-Trp-Trp-Trp) specifically antagonizes extracellular ANXA1 binding to its receptor FPR2 [41], we investigated their potential antiviral effects against PAstV. Treatment with ANXA1-ODN significantly reduced PAstV infection in a dose-dependent manner (≥80 nM) in IPEC-J2 cells, but had minimal effect in PK15 cells, indicating cell-specific protective effects (Fig 5F). However, treatment with WRW4 markedly inhibited PAstV replication in both IPEC-J2 and PK15 cells (Fig 5G). Collectively, these findings indicate that ANXA1 enhances PAstV-induced apoptosis through caspase-3 activation and promotes antiviral signaling via the RIG-I pathway, thus playing a pivotal role in modulating viral replication and host cell immune responses.

Ollectively, our data point to a two-step role for ANXA1 in PAstV infection. First, ANXA1 promotes viral attachment at the cell surface by engaging the acidic region of the ORF2 capsid via its R3 repeat. Second, ANXA1 amplifies RIG-I–IRF3 signalling to support IFN-β induction, as evidenced by blunted RIG-I upregulation and reduced IRF3 phosphorylation in ANXA1-deficient cells (Fig 6).

## Discussion

AstVs infect humans, many mammals, and birds and, in infants and young children, rank with rotavirus and norovirus among the leading causes of gastroenteritis [42]. Despite progress since their first description in 1975, key steps at the host–virus interface remain unclear. Work has lagged behind other enteric viruses and focused on the gut, yet extraintestinal disease is increasingly reported, including neurological, hepatic and renal involvement [43–46]. Advances are limited by difficulties in virus isolation, poor growth in culture, low titres, unstable infectivity and the lack of small-animal models. Porcine astrovirus shows broad cell tropism and mirrors key features of HAstV, making it a practical model [4,47]. A cellular factor that mediates PAstV entry has not been defined. Here we used a genome-wide CRISPR screen with PAstV-induced cytopathic effect as the readout to probe host requirements. Unlike a previous HAstV study that relied on CRISPR activation because cultured cells showed little cytopathology [9], our loss-of-function screen recovered several candidates and highlighted ANXA1 as an entry factor that supports PAstV uptake. We did not recover the HAstV factors FcRn or DPP4, consistent with species and cell-type differences in receptor use. Prior work suggests that astroviruses can rely on distinct combinations of protein receptors, cofactors and glycans that vary by clade [48,49].

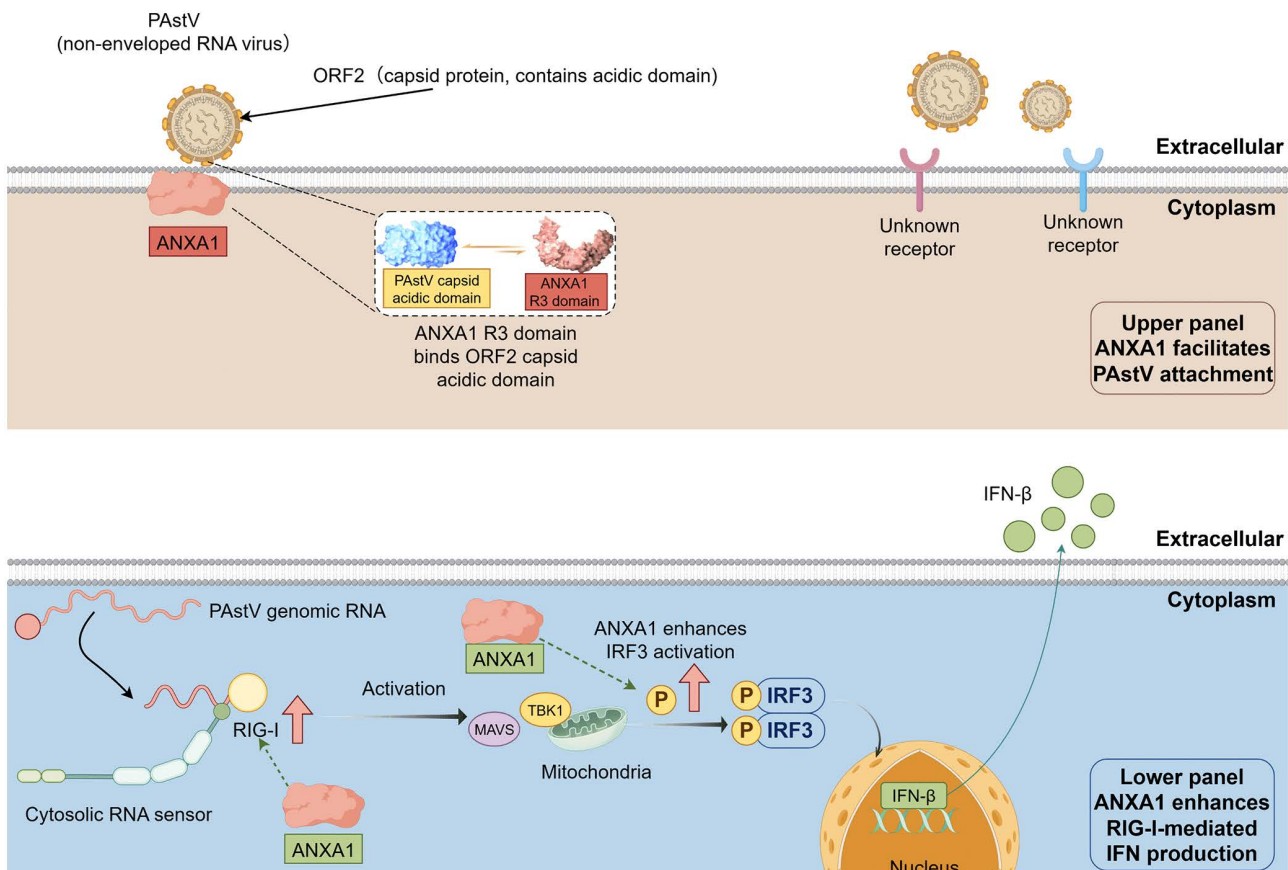

**Fig 6. A working model for the dual-stage roles of ANXA1 during PAstV infection.** Upper panel: At the cell surface, ANXA1 promotes PAstV attachment. The PAstV virion is depicted as a non-enveloped capsid. Inset, the R3 repeat of ANXA1 engages the acidic domain within the ORF2 capsid. Additional, as-yet-unidentified attachment factors or receptors may contribute to PAstV binding. Lower panel: Following infection, PAstV genomic RNA is sensed in the cytosol by RIG-I, triggering MAVS-dependent signalling and downstream activation of IRF3, culminating in induction of IFN-β transcription and type I interferon output. ANXA1 enhances this RIG-I–IRF3 axis. Dashed lines indicate proposed or indirect interactions supported by the data, and arrows indicate the direction of events. This figure was created with FigDraw (license code: YPWIR12153).

ANXA1 appears to act as an entry cofactor rather than a sole receptor for PAstV. Cells lacking ANXA1 show reduced viral binding, decreased early viral RNA and capsid signals, and slower propagation, whereas restoring or overexpressing ANXA1 enhances infection and an ANXA1-blocking antibody impairs virion attachment. Notably, when internalization is normalized to bound virus levels, uptake efficiency remains similar with or without ANXA1, implying that ANXA1's impact is mainly during initial attachment and uptake rather than later steps. In effect, ANXA1 increases the likelihood that a cell-bound virion will successfully enter and initiate productive infection, although the incomplete loss of infection in ANXA1-knockout cells shows that PAstV can still utilize other entry routes. This partial dependence on ANXA1 supports the idea that virus entry often involves multiple host factors, many viruses first engage low-affinity glycans to capture virions on the cell surface, then use protein receptors or co-receptors to trigger uptake [50]. PAstV likely follows this paradigm. ANXA1 may organize membrane entry sites and connect bound virions to the endocytic machinery, but additional elements are required for efficient entry. Candidate factors include cell-surface glycans like heparan sulfate as initial attachment factors, and membrane proteins that serve as entry receptors or triggers. In addition, lipid raft microdomains could cluster these receptors, and local proteases might be needed to activate the viral capsid [1]. Within this framework,

ANXA1 lowers the threshold for infection by enhancing early steps, but alone does not make a cell fully permissive. Clarifying ANXA1's precise place in the broader entry network will require further studies, such as genetic screens, glycan competition, proteomic searches for binding partners, and single-cell analyses linking entry steps to replication.

ANXA1 is abundantly expressed in diverse cell types [51]. Although primarily cytosolic, ANXA1 can translocate to the plasma membrane and associate with endosomal vesicles, these properties enable ANXA1 to support viral attachment [16,52]. However, PAstV tropism is not dictated by ANXA1 alone; productive entry also requires additional cell-surface and endocytic factors, include: glycans, lipid rafts, proteases & pH, innate barriers [53–56]. Together, these factors determine which cells permit efficient virus entry and replication. Differences in ANXA1 abundance, surface exposure, and trafficking likely tune susceptibility. Indeed, ANXA1 levels vary widely by tissue, altering the accessible pool for virus attachment. Cells with high ANXA1 at the apical surface or rapid endosomal recycling favor virus uptake, whereas cells where ANXA1 remains cytosolic or is downregulated limit entry despite the protein's presence. ANXA1 expression is dynamic – inflammatory or hormonal signals modulate its levels and differentiation alters its distribution, adding a temporal layer to tropism [51]. Blocking ANXA1 is expected to have the greatest impact in tissues where its accessible pool is high, whereas cells using alternative attachment factors may be less affected. A next step is to map ANXA1 levels and localization in target tissues alongside other potential co-factors.

The highly acidic C-terminal capsid domain varies markedly between astrovirus genotypes (the C-terminal half of the capsid shows only ~39–77% amino acid identity between serotypes, versus >84% identity in the N-terminal core) [57]. Such divergence could modulate reliance on host factors like ANXA1, meaning ANXA1-dependence might not be uniform across all astroviruses. Indeed, classical HAstV serotypes differ in receptor usage: the FcRn was recently identified as a common receptor for HAstV, with recombinant FcRn binding capsid spikes from serotypes 1, 4, and 8 [58], whereas HAstV-8 and related strains also utilize DPP4 as an entry cofactor [9]. Even subtle capsid differences can alter host factor engagement – for example, certain HAstV strains (HAstV-1, -8) require the disulfide isomerase PDIA4 for capsid disassembly, while HAstV-2 does not underscoring the need to assess ANXA1 across diverse astrovirus lineages. Structural studies of the astrovirus spike have identified conserved surface patches that likely mediate receptor binding [59], but surrounding variable regions may engender genotype-specific receptor or cofactor interactions. We therefore emphasize that further genotype-specific assays (spanning additional porcine astrovirus genotypes and other astrovirus species) are necessary to determine if ANXA1 dependence correlates with capsid architecture, thereby distinguishing shared versus lineage-specific entry mechanisms within the Astroviridae [60,61].

ANXA1 shows a dual action during astrovirus infection that helps reconcile our early and late readouts. At the start of infection, ANXA1 acts as an entry cofactor: its calcium-dependent binding to phospholipids supports virion attachment and endocytic routing, consistent with work in influenza A virus where ANXA1 enhances binding and delivery to endosomes and colocalizes with viral components [19,62]. In our study, ANXA1 knockout reduced PAstV binding and early RNA and protein, and slowed growth before 48 h, while overexpression and rescue increased susceptibility—evidence that ANXA1 raises the probability of productive entry. As infection progresses, ANXA1 also modulates innate signalling and cell fate. In epithelial systems, ANXA1 promotes the RIG-I–TBK1–IRF3–type I interferon axis and can prime apoptosis; our RNA-seq of ANXA1-deficient cells infected with PAstV showed altered expression of antiviral and apoptotic genes, indicating a dampened response. These time-dependent effects explain why early replication is lower in ANXA1-KO yet titres converge later: removal of an entry aid reduces initial seeding, but loss of ANXA1-driven antiviral signalling may partly relax later constraints [63]. A simple spatial model fits these observations: ANXA1 redistributes between the plasma membrane and endosomal compartments after infection, first facilitating capture and uptake, then shaping downstream signalling and cell death. Similar "two-phase" behaviour has been noted for other host factors that couple entry pathways to innate sensing [17,19].

ANXA1 has potential as a therapeutic target, but translation will require careful evaluation. Small-molecule antagonists such as WRW4 and antisense oligonucleotides targeting ANXA1 reduce infection in vitro. Yet ANXA1 also supports tissue

homeostasis and the resolution of inflammation via FPR2/ALX in the heart, platelets, and cerebral endothelium, so broad or prolonged inhibition may carry on-target safety risks [64–66]. Specificity is a further concern because agents in this class can engage other formyl peptide receptors, which may produce unwanted effects, Drug delivery, pharmacokinetics, and tissue penetration will also shape efficacy and safety [14,65]. In light of these constraints, and evidence that ANXA1 can promote RIG-I and interferon responses, we temper therapeutic claims and view ANXA1 as an entry cofactor that should be targeted with caution. Future work should define tissue- and compartment-specific roles during PAstV infection and prioritize strategies that limit systemic exposure, such as short-course or local delivery, or approaches that block the virus–ANXA1 interface rather than global ANXA1 signaling.

This study has limits that point to next steps. First, our evidence is from PK15 and IPEC-J2 cells in vitro. The in-vivo relevance remains to be tested, and robust porcine models for astrovirus are still difficult to establish. Second, the molecular interface between ANXA1 and the PAstV capsid is unresolved. Structure–function work with targeted mutations in annexin repeats and the acidic capsid domain, together with high-resolution structural methods, will be needed. Third, our strain panel is narrow, focused on genotypes 1 and 5. Broader testing across PAstV genotypes and representative astroviruses will clarify how general ANXA1 use is and whether dependence varies by lineage. Finally, ANXA1 likely has dual roles: it aids early entry and also shapes RIG-I signalling and apoptosis. Time-resolved experiments that match viral load between groups, alongside infection-independent stimulation of RIG-I pathways, will help separate entry effects from downstream signalling. These steps will refine where ANXA1 sits in the entry network and whether it is a tractable target.

Taken together, our study identifies ANXA1, which emerged from the screen, as a facilitator of entry. ANXA1 knock-out in PK15 lowered virus binding, early RNA and capsid signal, and delayed growth before 48 hours; overexpression and genetic rescue increased susceptibility. Antibody blocking reduced binding, while internalization per bound particle was similar across groups, indicating a role at attachment and the first steps of uptake. Biochemical assays and imaging showed that ANXA1 engages the acidic region of the capsid and colocalizes with incoming virus. Comparable reductions were seen in ANXA1-deficient IPEC-J2 cells, whereas unrelated RNA viruses showed no consistent change. During infection, ANXA1 levels rose, and transcriptome analysis indicated changes in RIG-I–IFN signalling and apoptosis. Infection was reduced but not abolished in the absence of ANXA1, consistent with additional factors contributing to entry. Together, these data identify ANXA1 as a host factor that promotes early PAstV entry and influences innate responses, refining the current view of astrovirus–host interaction.

## Methods

### Ethics statement

The ethical approval for this research was granted by the Institutional Animal Care and Use Committee (IACUC) of Guangxi University (Reference Number: GXU-2024–102). All methods were performed in accordance with the relevant guidelines and regulations.

### Cells and viruses

Porcine Kidney-15 (PK15, ATCC Cat# CCL-33), Human Embryonic Kidney 293T (HEK293T, ATCC Cat# CRL-3216), and Intestinal Porcine Epithelial Cell line-J2 (IPEC-J2, CVCL_2246) cells were cultured in Dulbecco's Modified Eagle Medium (DMEM, Gibco USA) supplemented with 10% fetal bovine serum (FBS, Gibco USA) and 1% penicillin-streptomycin at 37°C with 5% $CO_2$. All cell lines were confirmed mycoplasma-free.

PAstV-GX1 was isolated from fecal samples of diarrheic pigs and propagated in PK15 cells. The full genomic sequence of PAstV-GX1 was deposited in GenBank under accession number KF787112. A recombinant PAstV-GX1 expressing iLOV was previously constructed, validated, and maintained in our laboratory [35]. Additional viruses including GETV, EVG, SVV, MRV, BEV, and PAstV Type 5 were isolated, constructed, and maintained in our lab, with GenBank accession numbers MT269657, MT274669, MK039162, OQ627746-OQ627755, OL630964, and PP968017, respectively.

## Plasmids

Lentiviral vectors pLenti CMV GFP Hygro and lentiCRISPR v2, and packaging plasmids pMD2.G and psPAX2, were generously provided by Dr. Zuzhang Wei (Guangxi University). For CRISPR sgRNA constructs, paired oligonucleotides (50 µM each) were annealed and cloned into BbsI- or BsmBI-linearized lentiCRISPR v2 vectors (NEB, USA). The full-length cDNA sequences of ANXA1 and PAstV Capsid were cloned into pcDNA 3.1(+)-HA and pCAGGS-Flag vectors using HindIII/EcoRI or BamHI/NotI restriction enzymes, respectively. PAstV capsid-acidic domain, full-length ANXA1, and ANXA1 repeat-3 were cloned into pET-32a(+) and verified by sequencing.

## Antibodies and reagents

Primary antibodies used included anti-ANXA1 (21990–1-AP), anti-RIG-I (67556–1-Ig), anti-IRF3 (11312–1-AP), anti-phospho-IRF3 (29528–1-AP), anti-caspase 3/P17/P19 (82202–1-RR), anti-Flag (66008–4-Ig), anti-beta-tubulin (10094–1-AP), and anti-HA-HRP (HRP-81290) were purchased from Proteintech, China. Secondary antibodies, goat anti-mouse IgG Alexa Fluor 488 (ab150113) and goat anti-rabbit IgG Alexa Fluor 594 (ab150080), were purchased from Abcam (UK). Horseradish peroxidase (HRP)-conjugated anti-mouse and anti-rabbit antibodies (A0216, A0208) and DiI membrane staining kit (C1991S) were obtained from Beyotime (China). WRW4 (HY-P1119) was sourced from Sigma-Aldrich (USA). 5′ppp-RNA (5′ppp-dsRNA) was obtained from Invivogen (CA, USA) and used at working concentration of 1 µg/ml.

## Genome-scale CRISPR screening in PK15 cells

Approximately $6 \times 10^7$ PK15 cells from a genome-wide CRISPR mutant library were infected with PAstV-GX1 at an MOI of 0.01 in FBS-free DMEM. After 1-hour incubation at 37°C and 5% $CO_2$, inoculum was replaced with fresh DMEM supplemented with 0.5 µg/mL TPCK-treated trypsin (Sigma-Aldrich, Cat# 4370285) and 1% penicillin-streptomycin. Surviving cells were collected 4 days post-infection, expanded, and subjected to additional infection rounds at an MOI of 0.1. After four screening rounds, high-throughput sequencing was performed on surviving cells from rounds two and four to identify candidate genes.

## Generation of ANXA1 knockout cell line using CRISPR/Cas9

Single sgRNA targeting ANXA1 was cloned into the lentiCRISPR v2 vector, packaged into lentiviruses, and transduced into PK15 and IPEC-J2 cells. Cells were selected using 3 µg/mL puromycin, and single clones were isolated by limiting dilution, expanded, and validated by Sanger sequencing and Western blot analysis.

## Cell viability assay

Cell proliferation of WT and ANXA1-KO cells was assessed using the Cell Counting Kit-8 (CCK-8, G4103, Servicebio, China). Cells were seeded into 96-well plates, incubated for 36 hours, treated with CCK-8 reagent, and absorbance at 450 nm was measured using a microplate reader.

## Recombinant protein expression and antibody preparation

To produce recombinant ANXA1, ANXA1-R3, and PAstV-acidic proteins, the constructed plasmids and empty pET32a vector were transformed into Escherichia coli strain BL21(DE3) (TransGen Biotech, China). Cultures were grown at 37°C until reaching logarithmic growth phase, after which protein expression was induced with 0.5 mM isopropyl-β-D-thiogalactopyranoside (IPTG, Solabio, China), followed by additional incubation at 37°C. Recombinant proteins were purified using a His-Bind Kit (Novagen Inc., Germany). Purified proteins were then employed to immunize Kunming mice, generating anti-ANXA1 polyclonal antibodies (ANXA1-pAb). Sera from both immunized and non-immunized mice were dialyzed against binding buffer (20 mM sodium phosphate, pH 7.0) at 4°C for 24 hours, with three buffer changes, to

remove excess compounds. Additionally, ANXA1-pAb and normal rabbit IgG were purified from serum using a HiTrap Protein L column (GE Healthcare, USA), eluted with elution buffer (0.1 M glycine-HCl, pH 2.7), and immediately neutralized with 1 M Tris-HCl (pH 9.0).

### Transfection

Lipofectamine 3000 (L3000015, Invitrogen, USA) was employed according to manufacturer guidelines. Briefly, plasmids or siRNA were diluted in Opti-MEM, mixed with Lipofectamine reagent, and incubated for 20 minutes at room temperature before adding to cells. Media was replaced with fresh DMEM containing 10% FBS after 6 hours.

### Virus infection and titration

WT and ANXA1-KO cells were infected with PAstV at MOIs of 0.01 or 0.1. Supernatants were collected at designated times, serially diluted, and added in octuplicate to new cell cultures. Viral titers were calculated using the Reed-Muench method to determine $TCID_{50}$ at 72 hours post-infection.

### Western blot and immunoprecipitation

Proteins from cell lysates were separated by SDS-PAGE, transferred onto PVDF membranes, and probed with specific primary and HRP-conjugated secondary antibodies. Immunoprecipitation assays were performed as previously described, using Flag- or HA-tagged proteins expressed in HEK293T cells, captured with BeyoMag Protein A + G beads (Beyotime, China).

### Indirect immunofluorescence assay and confocal microscopy

Cells were infected with PAstV at indicated MOIs, fixed with 4% PFA, permeabilized, blocked, and incubated sequentially with primary and fluorescent-conjugated secondary antibodies. Nuclei were stained with DAPI, and images were captured on Thermo EVOS M5000 and Zeiss LSM880 confocal microscope.

### Absolute quantitative real-time PCR

Total RNA was extracted using RNAiso Plus (Takara, Japan), reverse-transcribed, and quantified by SYBR Green qPCR (Vazyme, China) using LightCycler 96 (Roche, Switzerland). Viral RNA copy number was determined using a standard curve based on a plasmid encoding ORF2.

### Duolink proximity ligation assay

Proximity ligation assay was performed in PK15 using the Duolink In Situ Orange kit (Sigma-Aldrich DUO92102). Cells were grown in 35-mm glass-bottom confocal dishes, infected with PAstV at MOI 1 and processed at 24 h post-infection. Cells were washed, fixed in 4% paraformaldehyde, permeabilized with 0.1% Triton X-100, and blocked with Duolink Blocking Solution. Rabbit anti-ANXA1 and mouse anti-Capsid were applied in Duolink Antibody Diluent, followed by anti-rabbit PLUS and anti-mouse MINUS probes, ligation, and rolling-circle amplification with the Orange detection reagent according to the manufacturer's protocol. Nuclei were counterstained with DAPI, and samples were imaged by confocal microscopy.

### Assessment of ANXA1-dependent PAstV attachment and internalization dynamics

WT and ANXA1-KO cells were infected at indicated MOIs at 4°C (binding assay) or 37°C (internalization assay). Post-incubation, cells were washed, treated with citrate buffer, and virus RNA was quantified via qRT-PCR to assess differences in binding and entry efficiencies.

In certain conditions, PAstV was pre-incubated with varying concentrations of soluble ANXA1 or bovine serum albumin (BSA), or target cells were pretreated with anti-ANXA1 antibodies for 1 hour at 37°C. Following treatment, the virus–protein or virus–antibody mixtures were applied to cells on ice for 1 hour to facilitate viral attachment. Unbound viral particles were removed by washing the cells three times with cold PBS. Total RNA was then extracted from cells using TRIzol reagent, and surface-bound PAstV RNA levels were quantified by quantitative RT-PCR as a proxy for viral binding efficiency.

### Antibody blocking and viral attachment assays

PK15 cells were cultured in 6-well plates or confocal microscopy dishes until reaching 60–70% confluency. Cells were incubated with varying concentrations of ANXA1-pAb (5, 10, 20, and 50 µg/mL) or mouse IgG (50 µg/mL, control) at 4°C for 1 hour. Subsequently, antibodies were replaced with PAstV (MOI = 10), followed by incubation at 4°C for an additional 1 hour. Cells were then extensively washed with citrate buffer. Cells in 6-well plates were harvested for RNA extraction and viral RNA quantification by qPCR to assess relative viral attachment efficiency. Cells cultured on confocal dishes were fixed and subjected to immunofluorescence microscopy to visualize viral attachment.

### Flow cytometry

Flow cytometry was used to quantify PAstV-GX1-iLOV infection, reporter expression and apoptosis. Cells were infected as indicated, harvested at the specified time points, gently detached with trypsin-EDTA, washed with cold PBS and resuspended in binding buffer. For direct reporter readout, iLOV fluorescence was recorded in the FITC channel, and the percentage of iLOV-positive cells was taken as the infection rate. For immunofluorescence-based detection of viral or host proteins, cells were fixed with 4% paraformaldehyde, permeabilized (0.1% saponin or 0.1% Triton X-100 in PBS), blocked in 1% BSA and incubated sequentially with primary antibodies and fluorophore-conjugated secondary antibodies; isotype and unstained samples were included as controls. For apoptosis analysis, an Annexin V-FITC/PI kit (Beyotime, C1062S, China) was used according to the manufacturer's instructions: briefly, cells were incubated with Annexin V-FITC and PI for 10–20 min at room temperature in the dark before acquisition. Data were acquired on a BD flow cytometer (BD FACS-Diva software) and analyzed with FlowJo (BD Biosciences) after exclusion of debris and doublets based on FSC/SSC and pulse geometry.

### Structural modeling and molecular docking analysis

To investigate the structural basis of the interaction between ANXA1 and the PAstV Capsid protein, we predicted their three-dimensional structures using AlphaFold3 based on the respective amino acid sequences. The top-ranked models, as determined by the highest pLDDT confidence scores, were selected as representative structures for subsequent analyses. Protein–protein docking was performed using the locally deployed HDOCKlite v1.1 platform. HDOCK integrates physics-based and bioinformatics-driven algorithms to conduct global sampling of potential interaction modes and ranks docking poses based on binding energy and confidence score. Given the absence of prior knowledge regarding the binding interface, a blind global docking approach was employed, exploring six degrees of freedom (three translational and three rotational parameters). o assess the dynamic stability of the ANXA1–Capsid complex, molecular dynamics (MD) simulations were conducted using the AMBER22 software suite. The AMBER ff14SB force field was applied for protein parametrization. The complex was solvated in a truncated octahedral box of TIP3P water molecules with a 1.2 nm buffer and neutralized with counterions. After energy minimization and equilibration at 300 K and 1 bar, production runs were carried out for 100 ns. Trajectories were saved every 10 ps, yielding a total of 10,000 snapshots. Trajectory analyses were performed using the CPPTRAJ module, and binding free energies were calculated with the MMPBSA.py script.

## SPR Analysis of ANXA1–PAstV Capsid Binding Kinetics

SPR binding analysis was performed on a CM5 sensor chip using the following procedure: The chip surface was activated by injecting a freshly prepared mixture of 400 mM EDC and 100 mM NHS at 10 µL/min for 420 s. ANXA1-Repat3, diluted to 20 µg/mL in immobilization buffer, was then immobilized onto flow cell 2 (Fc2) at 10 µL/min, achieving a typical response level of ~12,600 RU; flow cell 1 (Fc1) served as an unmodified reference surface. Residual active esters were deactivated with a 420-s injection of 1 M ethanolamine hydrochloride (pH 8.5) at 10 µL/min. For binding kinetics, eight concentrations (0.02–1 µM) of PAstV-acidic in analyte buffer were serially injected in ascending order over both flow cells at 20 µL/min, with a 100-s association phase followed by a 180-s dissociation phase. Between each analyte cycle, the surface was regenerated. All results were analysed using BIAevaluation v.3.1 (GE Healthcare).

## RNA-seq and data analysis

RNA libraries were constructed from ANXA1-KO cells (PK15-ANXA1KO, PK15-ANXA1KO+PAstV) and WT cells (PK15-WT, PK15-WT+PAstV), with two biological replicates per group. Poly(A)+ RNA isolation, library preparation, and sequencing were conducted using the MGISEQ-2000RS platform (Shanghai Personal Biotechnology Co, Ltd, China). Sequence quality assessment was performed using FastQC (v0.11.7), and quality trimming was carried out with FASTX-Toolkit (v0.0.14), removing bases with Phred33 scores below 30 and retaining reads with lengths of at least 50 bp. The filtered reads were aligned to the Sus scrofa reference genome (v11.1) using HISAT2 (v2.1.0). Gene expression levels were quantified based on read counts using SAMtools (v1.7) and HTSeq-count (v0.9.1), expressed as fragments per kilobase of exon per million mapped fragments (FPKM). Differentially expressed genes (DEGs) were identified using DESeq2 (v1.30.1) with statistical significance set at $p \leq 0.05$ and fold-change $\geq 1$. Protein-protein interaction (PPI) networks for DEGs were constructed using STRING (https://string-db.org/), and Gene Set Enrichment Analysis (GSEA) was subsequently performed. Visualization of selected DEGs in PPI networks was conducted using Cytoscape software.

## Functional inhibition of ANXA1 and PAstV infection assay

To determine the functional role of ANXA1 in PAstV infection, we performed pharmacological inhibition using the small-molecule antagonist peptide WRW4 and antisense ODN targeting ANXA1 in PK15 and IPEC-J2 cells. WRW4, an antagonist of the FPR2/ANXA1 axis, was tested at concentrations of 0.3, 5, 8, and 10 µM. For ODN-mediated knockdown, ANXA1-targeting ODN were transfected at concentrations of 0.2, 0.3, 0.5, 0.8, and 1.0 nM using Lipofectamine 3000 (Invitrogen, USA). Cells were pre-treated with inhibitors for 24 hours at 37°C in a 5% $CO_2$ incubator prior to viral infection. Following pre-treatment, cells were infected with PAstV at MOI of 0.01 and incubated for an additional 36 hours. Viral infection efficiency was assessed by measuring PAstV ORF2 gene expression levels via qRT-PCR. Each experimental condition was performed with at least three biological replicates to ensure reproducibility and reliability.

## Statistical analysis

All experiments were performed with $n \geq 2$ biological replicates (as indicated). All the statistical analyses were performed with GraphPad Prism 9.5.1 software. The significance of differences was analyzed with Student's t-test or Two-way ANOVA test. The data were presented as means ± SD. ns, $P > 0.05$; *$P < 0.05$; **$P < 0.01$; ***$P < 0.001$.

Fiji 9.3.1 was used for densitometry of Western blots and for fluorescence quantification. Background was subtracted, ROIs were kept the same across samples, and values were normalized to the loading control or to cell area. For colocalization, Pearson's r and Manders coefficients were obtained with Coloc2 under consistent thresholds.

## Supporting information

**S1 Fig. CRISPR screening identifies ANXA1 as a entry factor for PAstV infection.** (A), Multi-step growth curves of PAstV-infected PK15 and IPEC-J2 cells. (B), Cytopathic effects in PK15 cells infected with different MOIs of PAstV. (C), RT-qPCR validation of gene knockdown efficiency using top-ranked enriched sgRNAs or non-targeting control sgRNA. (D), Flow cytometry of PAstV capsid staining in siRNA-transfected PK15 cells, showing the percentage of PAstV$^+$ cells for each targeted gene compared with NT. Data represent mean ± SD (n = 3). Statistical analysis was performed by unpaired Two-way ANOVA. (ns, P > 0.05; *P < 0.05; **P < 0.01; ***P < 0.001). (DOCX)

**S2 Fig. ANXA1 supports PAstV infection and facilitates early entry, and rescue restores susceptibility.** (A) GFP-positive plasmids used as transfection controls to assess sgRNA packaging efficiency. (B) Sequencing confirmation of ANXA1 knockout (PK15-ANXA1KO) compared to WT cells. (C) Cytopathic effects in PK15-WT and PK15-ANXA1KO cells infected with PAstV (MOI = 1) for 24 h. (D) Virus internalization in PK15 WT and ANXA1 KO. Cells were bound with PAstV at 4°C for 60 min, washed, shifted to 37°C for 30 min, surface virus removed by trypsin, and PAstV genomes quantified by RT-qPCR. Readouts: internalized genomes per well and internalization efficiency, defined as internalized divided by bound. (E) Sequencing result of the ANXA1 rescue plasmid. (F) Preparation and expression of recombinant ANXA1 protein; 1: induced recombinant protein; 2: induced pET-32a vector; 3: non-induced recombinant protein; 4: non-induced pET-32a vector; 5: purified supernatant; 6: purified precipitated protein. (G) Immunofluorescence analysis of PAstV infection (MOI = 0.01) in IPEC-J2-WT and IPEC-J2-ANXA1KO polyclonal knockout cells. Data represent mean ± SD (n = 3). Statistical analysis was performed by unpaired Two-way ANOVA. (ns, P > 0.05; *P < 0.05; **P < 0.01; ***P < 0.001). (DOCX)

**S3 Fig. ANXA1 directly binds to the PAstV ORF2 acidic domain via its repeat III domain.** (A) Western blot analysis confirming expression of various viral protein constructs in HEK-293T cells. (B) BiFC analysis showing interaction between ANXA1 and ORF2 plasmids. (C) In silico prediction of interaction between ANXA1 and ORF2 proteins. (D) Homology analysis of ANXA1 among porcine, human, and mouse species. (E) PCR results for recombinant ANXA1 plasmid construction using SOE technique. (F) Flow cytometry analysis of PAstV-ILOV infection (MOI = 1) in PK15-ANXA1KO cells transfected with various ANXA1 recombinant plasmids. (DOCX)

**S4 Fig. ANXA1 modulates virus-induced apoptosis and RIG-I–IRF3 signalling.** (A) BEV RNA loads at 24 hpi in PK15-WT and PK15-ANXA1KO cells (RT–qPCR, left) and Annexin V/PI staining of early apoptotic cells (Annexin V$^+$PI$^-$, right). (B) Gene expression distribution for each RNA-seq sample. (C) Sample-to-sample correlation heatmap. (D) PCA of transcriptomes. E, Summary of DEGs. (F) Volcano plot of DEGs between PAstV-infected PK15-WT and PK15-ANXA1KO cells. (G) GO enrichment for selected DEGs. H, WB of p-IRF3 in PK15-WT and PK15-ANXA1KO cells transfected with 5′ppp-dsRNA. (DOCX)

**S1 Table. Kinetic analysis of the interactions between ANXA1-Repat3 and the PAstV acidic domain.** (XLSX)

**S2 Table. Quality assessment of the RNA sequencing dat.** (XLSX)

**S3 Table. Summary statistics of cleaned reads mapped to the reference genome.** (XLSX)

**S1 Data. Underlying data for all figures and tables.** This file contains the numerical values underlying all plots and summary statistics reported in the main and supplementary figures and tables.
(XLSX)

## Author contributions

**Conceptualization:** Yuhang Luo, Qingting Dong, Shiqin Yi, Yifeng Qin, Weijian Huang.

**Data curation:** Yuhang Luo, Qingting Dong, Yifeng Qin.

**Formal analysis:** Yuhang Luo, Wenchao Zhang.

**Funding acquisition:** Yuhang Luo, Qingting Dong, Weijian Huang.

**Investigation:** Yuhang Luo, Qingli Fang, Wenchao Zhang, Ying Chen, Yeshi Yin.

**Methodology:** Yuhang Luo, Qingting Dong, Shiqin Yi, Wenting Zhang, Qingli Fang, Wenchao Zhang, Ying Chen, Yeshi Yin, Weijian Huang.

**Project administration:** Yuhang Luo, Qingting Dong, Shiqin Yi, Wenting Zhang, Wenchao Zhang, Yifeng Qin, Weijian Huang.

**Resources:** Yuhang Luo, Wenting Zhang, Yiyang Du, Zuzhang Wei, Yifeng Qin.

**Software:** Yuhang Luo, Wenting Zhang, Yiyang Du, Yeshi Yin, Yifeng Qin.

**Supervision:** Shiqin Yi, Yiyang Du, Kang Ouyang, Ying Chen, Zuzhang Wei, Yifeng Qin.

**Validation:** Shiqin Yi, Yiyang Du, Qingli Fang, Kang Ouyang, Ying Chen, Zuzhang Wei.

**Visualization:** Yiyang Du, Qingli Fang, Kang Ouyang, Zuzhang Wei, Weijian Huang.

**Writing – original draft:** Yuhang Luo.

**Writing – review & editing:** Kang Ouyang, Ying Chen, Zuzhang Wei, Yifeng Qin, Weijian Huang.

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
