## [Decision Letter · Decision Letter 0]

21 Oct 2025

PPATHOGENS-D-25-02239

Genome-wide CRISPR screening identifies Annexin A1 is a critical host receptor facilitating porcine astrovirus entry and replication

PLOS Pathogens

Dear Dr. Huang,

Thank you for submitting your manuscript to PLOS Pathogens. After careful consideration, we feel that it has merit but does not fully meet PLOS Pathogens's publication criteria as it currently stands. Therefore, we invite you to submit a revised version of the manuscript that addresses the points raised during the review process.

Please submit your revised manuscript within 60 days Dec 20 2025 11:59PM. If you will need more time than this to complete your revisions, please reply to this message or contact the journal office at plospathogens@plos.org. Please include the following items when submitting your revised manuscript:

We look forward to receiving your revised manuscript.

Kind regards,

Roberto Cattaneo

Guest Editor

PLOS Pathogens

Sonja Best

Section Editor

PLOS Pathogens
Sumita Bhaduri-McIntosh

Editor-in-Chief

PLOS Pathogens

orcid.org/0000-0003-2946-9497
 
Michael Malim
Editor-in-Chief

PLOS Pathogens

orcid.org/0000-0002-7699-2064

**Additional Editor Comments:**

All reviewers suggested toning down the title and the short title, and providing a more balanced discussion. Considering the strength of the data presented, I suggest to change the title to something like "Genome-wide CRISPR screening identifies Annexin A1 as a facilitator of astrovirus entry" ("critical receptor" and "replication" should be deleted). The discussion needs quite some work; the five questions asked by reviewer 1 can serve as blueprint to improve the discussion. A 2024 review on "Cellular receptors for mammalian viruses" published by PLOS pathogens may also help putting these results in the context of current knowledge.

**Journal Requirements:**

At this stage, the following Authors/Authors require contributions: Weijian Huang. Please ensure that the full contributions of each author are acknowledged in the "Add/Edit/Remove Authors" section of our submission form.

https://journals.plos.org/plospathogens/s/submission-guidelines#loc-parts-of-a-submission

- ® on pages: 26, and 29

- TM on pages: 25, and 29.

5) We have noticed that you have uploaded Supporting Information files, but you have not included a list of legends. Please add a full list of legends for your Supporting Information files after the references list.

Potential Copyright Issues:

i) We note that  Figures 1, 2, and 6 are created through BioRender. Please confirm that you hold a Premium account and provide a pdf copy of the CC BY 4.0 Licence as provided by BioRender. For instructions on how to generate a CC BY 4.0 license for your figure, please see the guidelines here: https://help.biorender.com/hc/en-gb/articles/21282341238045-Publishing-in-open-access-resources.

If you are using the free assets from BioRender, we are unable to publish these images as they are licenced under a stricter licence than CC BY 4.0. In this case we ask you to remove the BioRender images and replace them with open source alternatives.

See these open source resources you may use to replace images / clip-art:

- https://bioart.niaid.nih.gov/

- https://bioicons.com/

- https://healthicons.org/

- https://scidraw.io/

- https://reactome.org/icon-lib

- https://www.phylopic.org/images

- https://journals.plos.org/plosbiology/article?id=10.1371/journal.pbio.3002395

7) Please amend your detailed Financial Disclosure statement. This is published with the article. It must therefore be completed in full sentences and contain the exact wording you wish to be published.

8) Kindly revise your competing statement in the online submission form to align with the journal's style guidelines: 'The authors declare that there are no competing interests.'

**Reviewers' Comments:**

Reviewer's Responses to Questions

**Part I - Summary**

Reviewer #1: Luo and colleagues identify and characterize the critical host receptor facilitating porcine astrovirus (PAstV) entry and replication, using genome-wide CRISPR-Cas9 knockout screening in porcine epithelial cells. The main finding is that Annexin A1 (ANXA1) is necessary for for PAstV entry, as confirmed by genetic ablation, pharmacological inhibition, and functional rescue studies. Mechanistic assays demonstrated direct, high-affinity binding between ANXA1 and the acidic C-terminal domain of the PAstV ORF2 capsid, with ANXA1 mediating both viral attachment/internalization and innate antiviral signaling through the RIG-I pathway. ANXA1’s role was shown to be selective for PAstV, not affecting unrelated RNA viruses. Pharmacological ANXA1 antagonists robustly suppressed PAstV replication, suggesting therapeutic potential. The story is of interest and well-performed, including mechanistic exploration: molecular docking, protein-protein interaction, RNA-seq, apoptosis/cell death, and innate immune signaling. Some of the results are over-interpreted.

Reviewer #2: The manuscript by Luo et al. investigates host dependency factors for porcine astrovirus (PAstV) type 1 (and type 5) using a positive selection CRISPR screen in PK15 cells. They identify Annexin A1 (ANXA1) as a candidate for supporting PAstV entry and perform knockout, overexpression, and binding studies to evaluate its role. While there appears to be a lot of effort given to produce these data, and the potential for novel findings for this understudied pathogen, it was quite difficult to assess the quality of the data and whether it supports the main conclusions. For example, there were inconsistencies between the main text and figures, and a significant lack of experimental detail and methods (e.g. primer sequences). Even so, there was a lack of data to support the role for ANXA1 in PAstV replication, which was a main conclusion, as a more straightforward explanation for the data had not been considered (i.e. the loss of ANXA1reduces infectivity and therefore downstream responses like cell death and p-IRF3). A more comprehensive list of these challenges is noted below:

Reviewer #3: In this manuscript, Luo et al use a genome-wide CRISPR screen to identify cellular receptors for porcine astrovirus strain PAstV-GX1. Using a variety of techniques, the authors data suggests that annexin A1 may be an important binding partner for PAstV infection in PK15 cells. They go on to show that ANXA1 is involved in viral entry and replication, identify the viral-protein binding sites, the functional role of ANXA1 in modulating apoptosis and antiviral signaling during infection, and finally that pharmacologically targeting ANXA1 impairs PAstV infection. Overall, the studies are comprehensive and rigorous supported by good statistical analysis. The information is important for the astrovirus field. However, there are key issues that should be addressed including poor figure quality making it challenging to review the data even using the online images.

**Part II – Major Issues: Key Experiments Required for Acceptance**

Reviewer #1: 1. My main concern is the fact that ANXA1 Ko cells are still sensitive to infection, with about 50% decrease of sensitivity, and a delay in viral kinetics (Fig.2). ANXA1 is not a “critical receptor” as stated in the short title.

Reviewer #2: 1. Supp Fig 1; Line 108 - Missing statistical testing to support differences in virus titer between PK15 and IPEC-J2 cells. Also, differences in CPE are not shown in Supp Fig 1b, as only PK15 cells are shown, so the text should be clarified. Also, Supp Fig 1b is lacking uninfected control images to compare with virus infection. CPE is not very clear at MOI of 0.01 at 72 hours and the text should be clarified (lines 108-110).

2. Fig 1b lacks description in the Results section.

3. Lines 127-128 do not match the gene list in Supp Fig 1c so it is unclear which to follow or which is accurate.

4. Supp Fig 2c - the contrast is different in both transmitted light images, making it difficult to evaluate differences between the WT vs KO infected cells.

5. Fig 2d - at what hpi is the titer assessed in KO and WT cells? Same question for time point information in Fig 2F. This is a key piece of information to evaluate and determine if they support the conclusions.

6. Fig 2e - there is no difference in virus genome after 48 hours in the WT vs KO cells, how do the authors interpret these data?

7. Fig Supp 2d shows no significant difference in internalization, but wouldn’t it be expected to be different given that binding is reduced in the KO cells (so fewer viruses would enter)?

8. Fig Supp 2F shows purified ANXA1 protein, not the data described in lines 167-171 (competitive binding assay).

9. Figure 3G is not accurately depicting DAPI and merged images (some are just transmitted light + GFP). Or GFP only in the Merge column.

10. Fig. 3h lacks description in the Results section.

11. Can the authors speculate on why ANXA1 expression goes up during infection? Typically receptor expression will go down to prevent superinfection.

12. Can the authors explain the 2 ANXA1 bands present in the WT-NTC and A1KO-rescue cells in Fig 3d? In the above overexpression (Fig 3a), only a single ANXA1 band is present.

14. Fig 4b examines the correlation of signal between transiently expressed ANXA1 and ORF2 (I think a more appropriate label would be “Capsid”), but it is unclear how they are associating with one another. Wouldn’t the ANXA1 be expressed on the cell surface?

14. Fig 4d uses a proximity ligation assay between ANXA1 and ORF2 during infection, but the methods are completely lacking and it is unclear how this experiment was performed and how the images were acquired.

15. Lines 298-299 and Fig 5b - the authors conclude that loss of ANXA1 results in lower cell death, but isn’t it more likely that this lower cell death is due to reduced infection? The same explanation is not considered for Fig 5e, showing reduced p-IRF3

16. Figure 6 depiction of the virus is inaccurate - Astroviruses are non-enveloped but his image shows PAstV enveloped and budding from the cell surface.

17. There’s a complete lack of discussion about the likelihood of other entry factors. Given that there was not a complete loss of infectivity in either cell line used in the study, it is likely there are other factors involved.

Reviewer #3: 1. The authors show that ANXA1 KO cells have reduced viral genomic copies at 24 hr, yet Figure 2E shows that the genomic load is equivalent by 48 hr. This suggests that ANXA1 may not be the only receptor/binding partner. This is further supported by the data on Fig 2K. The authors must either confirm their interpretation that ANXA1 is the receptor or tone the language.

**Part III – Minor Issues: Editorial and Data Presentation Modifications**

Reviewer #1: 2. Are there tissue/cell type variations in ANXA1 expression that might dictate tropism or restrict antiviral efficacy? This could be discussed.

3. What is known about tissue-specific receptor expression and could be the impact on viral spread/pathology? This could be further discussed.

4. Are there expected genotype-dependent differences in receptor usage among the PAstV clades not tested here?

5. How could the different roles of ANXA1 (entry and immune signaling, apoptosis) influence net pathogenesis and replication?

6. Could ANXA1 be a receptor for other astroviruses, or is its role limited to PAstV?

Reviewer #2: Please see combined sections above

Reviewer #3: 1. Please discuss the tropism of ANXA1 - which cells express it and what is the subcellular location. If it is widely expressed, the authors need to discuss why PAstV tropism is limited.

2. The figure quality must be improved. The fonts are too small to read and some images, for example 1C, are indecipherable as presented.

3. The authors should describe the selection criteria used to identify the top candidates from the CRISPR screen. Including a representative image of the capsid staining used to quantitate viral replication is needed.

4. Throughout the text, the authors should define abbreviations. For example, the genes on lines 127 - 132.

5. In Figure 3G, the contrast of the SVV-GFP images appear to be significantly different. Please correct.

PLOS authors have the option to publish the peer review history of their article (what does this mean?). If published, this will include your full peer review and any attached files.

Reviewer #1: No

Reviewer #2: No

Reviewer #3: No

**Figure resubmission:**

After uploading your figures to PLOS’s NAAS tool - https://ngplosjournals.pagemajik.ai/artanalysis, NAAS will process the files provided and display the results in the "Uploaded Files" section of the page as the processing is complete. If the uploaded figures meet our requirements (or NAAS is able to fix the files to meet our requirements), the figure will be marked as "fixed" above. If NAAS is unable to fix the files, a red "failed" label will appear above. When NAAS has confirmed that the figure files meet our requirements, please download the file via the download option, and include these NAAS processed figure files when submitting your revised manuscript.
---

## [Decision Letter · Decision Letter 1]

18 Dec 2025

PPATHOGENS-D-25-02239R1

Genome-wide CRISPR screening identifies Annexin A1 as a facilitator of porcine astrovirus entry

PLOS Pathogens

Dear Dr. Huang,

Thank you for submitting your manuscript to PLOS Pathogens. After careful consideration, we feel that it has merit but does not fully meet PLOS Pathogens's publication criteria as it currently stands. Therefore, we invite you to submit a revised version of the manuscript that addresses the points raised during the review process.

We look forward to receiving your revised manuscript.

Kind regards,

Roberto Cattaneo

Guest Editor

PLOS Pathogens

Sonja Best

Section Editor

PLOS Pathogens

Sumita Bhaduri-McIntosh

Editor-in-Chief

PLOS Pathogens

orcid.org/0000-0003-2946-9497

Michael Malim

Editor-in-Chief

PLOS Pathogens

orcid.org/0000-0002-7699-2064

**Additional Editor Comments:**

As requested by the reviewers, the titles of 3 Figures should be corrected. The misleading graphic of Figure 6 should be re-drawn as instructed.

**Journal Requirements:**

**Reviewers' Comments:**

Reviewer's Responses to Questions

**Part I - Summary**

Reviewer #1: the authors have addressed my concerns. The titles of the figures remain misleading. ANX1 is not "crucial" f(fig 1) nor "required and sufficient" (Fig 2 and 3) for viral replication

Reviewer #2: The authors have addressed the majority of my comments and suggestions.

**Part II – Major Issues: Key Experiments Required for Acceptance**

Reviewer #1: (No Response)

Reviewer #2: Figure 6 still depicts an enveloped virus to represent PAstV, a nonenveloped virus, and this should be corrected. Only the icosahedral capsid with ssRNA genome (no outer envelope) should be shown for the virion entering the cell and it should not be fusing with the membrane.

**Part III – Minor Issues: Editorial and Data Presentation Modifications**

Reviewer #1: (No Response)

Reviewer #2: (No Response)

PLOS authors have the option to publish the peer review history of their article (what does this mean?). If published, this will include your full peer review and any attached files.

Reviewer #1: No

Reviewer #2: No

**Figure resubmission:**
---

## [Editor Report · Decision Letter 2]

26 Jan 2026

Dear Dr Huang,

We are pleased to inform you that your manuscript 'Genome-wide CRISPR screening identifies Annexin A1 as a facilitator of porcine astrovirus entry' has been provisionally accepted for publication in PLOS Pathogens.

Best regards,

Roberto Cattaneo

Guest Editor

PLOS Pathogens

Sonja Best

Section Editor

PLOS Pathogens

Sumita Bhaduri-McIntosh

Editor-in-Chief

PLOS Pathogens

orcid.org/0000-0003-2946-9497

Michael Malim

Editor-in-Chief

PLOS Pathogens

orcid.org/0000-0002-7699-2064

Figure 6 and its legend have been updated as instructed.

The manuscript can be accepted.
---

## [Editor Report · Acceptance letter]

Dear Dr Huang,

We are delighted to inform you that your manuscript, "Genome-wide CRISPR screening identifies Annexin A1 as a facilitator of porcine astrovirus entry," has been formally accepted for publication in PLOS Pathogens.

Best regards,

Sumita Bhaduri-McIntosh

Editor-in-Chief

PLOS Pathogens

orcid.org/0000-0003-2946-9497

Michael Malim

Editor-in-Chief

PLOS Pathogens

orcid.org/0000-0002-7699-2064